# Adult Neurogenesis of the Medial Geniculate Body: In Vitro and Molecular Genetic Analyses Reflect the Neural Stem Cell Capacity of the Rat Auditory Thalamus over Time

**DOI:** 10.3390/ijms25052623

**Published:** 2024-02-23

**Authors:** Jonas Engert, Bjoern Spahn, Sabine Sommerer, Totta Ehret Kasemo, Stephan Hackenberg, Kristen Rak, Johannes Voelker

**Affiliations:** Department of Otorhinolaryngology, Plastic, Aesthetic and Reconstructive Head and Neck Surgery, University Hospital Wuerzburg, Josef-Schneider-Strasse 11, 97080 Wuerzburg, Germany; spahn_b@ukw.de (B.S.); sabine.sommerer@stud-mail.uni-wuerzburg.de (S.S.); ehret_t@ukw.de (T.E.K.); hackenberg_s@ukw.de (S.H.); rak_k@ukw.de (K.R.); voelker_j@ukw.de (J.V.)

**Keywords:** auditory thalamus, neural stem cells, mRNA abundance, adult neurogenesis

## Abstract

Neural stem cells (NSCs) have been recently identified in the neonatal rat medial geniculate body (MGB). NSCs are characterized by three cardinal features: mitotic self-renewal, formation of progenitors, and differentiation into all neuroectodermal cell lineages. NSCs and the molecular factors affecting them are particularly interesting, as they present a potential target for treating neurologically based hearing disorders. It is unclear whether an NSC niche exists in the rat MGB up to the adult stage and which neurogenic factors are essential during maturation. The rat MGB was examined on postnatal days 8, 12, and 16, and at the adult stadium. The cardinal features of NSCs were detected in MGB cells of all age groups examined by neurosphere, passage, and differentiation assays. In addition, real-time quantitative polymerase chain reaction arrays were used to compare the mRNA levels of 84 genes relevant to NSCs and neurogenesis. In summary, cells of the MGB display the cardinal features of NSCs up to the adult stage with a decreasing NSC potential over time. Neurogenic factors with high importance for MGB neurogenesis were identified on the mRNA level. These findings should contribute to a better understanding of MGB neurogenesis and its regenerative capacity.

## 1. Introduction

Worldwide, around 700 million people will suffer from hearing loss requiring rehabilitation by 2050 [1]. Patients with sensorineural hearing loss (SNHL), which is characterized by an irreversible loss of function of cochlear and neuronal structures, can receive sensory rehabilitation through conventional and implantable hearing aids [2]. The causes of SNHL can be genetic, infectious, tumorous, and environmental [3]. In addition, cochlear deafness consecutively leads to pathological changes and loss of neuronal structures of the auditory pathway [4,5]. Conventional and implantable hearing aids are able to rehabilitate a certain degree of auditory function in patients with SNHL; however, these treatments do not regenerate lost cells [2,6].

In contrast, neural stem cells (NSCs) are a potential source for regeneration [7,8]. In an Alzheimer’s mouse model, the transplantation of NSCs significantly improved learning and memory function and a decrease in pathological plaque accumulation was observed [9]. NSCs are characterized by three cardinal features: the ability of mitotic self-renewal; formation of progenitor cells; and differentiation into all cell types of the neuroectodermal cell lineage [10]. The advantage of endogenous adult NSCs is that initiating regenerative processes does not depend on transplanting allogeneic cells [11,12]. The detection of an NSC niche was initially observed in two main areas of the adult brain: the subventricular zone and the dentate gyrus of the hippocampus [13,14]. Interestingly, NSC niches were demonstrated in nuclei of the auditory pathway up to the adult stage. The characteristics of NSCs were detected in cells of the adult spiral ganglion, cochlear nucleus, inferior colliculus, and auditory cortex [15,16,17,18]. The adult stem cell potential of the different parts of the auditory system differs. In in vitro analyses, a lower or no stem cell capacity was found in the adult cochlear and the adult cochlear nucleus compared to the adult vestibular organs or the adult inferior colliculus [15,17,19].

In addition, an NSC niche was discovered in the medial geniculate body (MGB) of the early postnatal rat [20]. The MGB is part of the thalamus, consists of three subnuclei, and is a small elevation on the lower edge of the surface of the thalamus between the lateral geniculate body and the pulvinar [21]. The ventral subnucleus receives afferent input from the inferior colliculus and projects to the auditory cortex. The dorsal nucleus transmits afferents to the auditory association cortex. The medial nucleus plays a vital role in the duration of sound and the intensity perception of sound [22,23]. Interestingly, it has recently been shown for the first time that the human MGB is surgically accessible. As part of deep brain stimulation to treat severe tinnitus, stereotactic surgery was performed on the human MGB [24]. The application of substances into the inner ear via a surgically inserted catheter already represents a promising translational approach for the treatment of cochlear hearing disorders [25]. Therefore, a better understanding of the regenerative capacity and neurogenesis of the MGB is of great interest. A better understanding of the neurogenic and molecular processes that influence the maturation of the MGB could provide the basis for future regenerative interventions on the MGB.

Insights about whether a neurogenic potential exists in the MGB up to the adult stage and which molecular genetic dynamics affect this potential represent the basis for one possible strategy to regenerate neuronal structures in the MGB after cochlear or neuronal pathologies of the auditory pathway. Therefore, neurosphere assays, differentiation studies, and immunocytological investigations were performed on postnatal days (PND) 8, 12, and 16, and at the adult stadium (PND 48). In addition, the mRNA abundance of neurogenic factors was analyzed.

## 2. Results

### 2.1. Cells of the MGB Form Neural Stem and Progenitor Cell Marker Positive Neurospheres with Decreasing Proliferative Capacity up to the Adult Stage

Free-floating and spherical cell conglomerates were developed in cell cultures of all age groups. The number of these primary neurospheres was evaluated at 100,000 cells seeded per cell culture (Figure 1). The data followed a Gaussian normal distribution; therefore, the mean and standard error of the mean (SEM) are shown. At PND 8 1035 ± 60.45, at PND 12 1135 ± 101, at PND 16 450 ± 72.09, and at PND 48 197 ± 13.49, neurospheres were detected (n = 6). A significant decrease in the number of neurospheres per 100,000 seeded cells was observed between PND 8 and PND 16 (*p* < 0.001), and PND 8 and PND 48 (*p* < 0.001), as well as between PND 12 and PND 16 (*p* < 0.001), and PN 12 and PND 48 (*p* < 0.001) (Figure 1). There were no significant differences between PND 8 and PND 12 (*p* = 0.7429) or between PND 16 and PND 48 (*p* = 0.0783) (Figure 1). Between adjacent ages, the most significant decrease was between PND 12 and PND 16, at 39.64% (Figure 1).

In addition, the formation of neurospheres per animal was analyzed (n = 6). Secondary, tertiary, and quaternary neurospheres were formed after passaging (P1, P2, P3) (Figure 2). A significant increase in the number of neurospheres was detected between the primary culture (PC) and the quaternary passage at all age groups examined (PC vs. P3: PND 8, *p* < 0.001; PND 12, *p* < 0.001; PND 16, *p* < 0.001; PND 48, *p* < 0.001) (Figure 2a–d). At PND 8, the increase in the number of neurospheres was significant between each passage (PC vs. P1, *p* = 0.0083; PC vs. P2, *p* < 0.001; PC vs. P3, *p* < 0.001; P1 vs. P2, *p* = 0.0164; P1 vs. P3, *p* < 0.001; P2 vs. P3, *p* < 0.001) (Figure 2a). A significant increase was detected at PND 12 between all passages, except between P2 and P3 (PC vs. P1, *p* < 0.001; PC vs. P2, *p* < 0.001; PC vs. P3, *p* < 0.001; P1 vs. P2, *p* = 0.0473; P1 vs. P3, *p* = 0.0316; P2 vs. P3, *p* = 0.9974) (Figure 2b). Interestingly, a significant increase in the number of neurospheres was not detected at PND 16 until P3 (PC vs. P1, *p* > 0.9999; PC vs. P2, *p* = 0.1517; PC vs. P3, *p* < 0.001; P1 vs. P2, *p* = 0.1362; P1 vs. P3, *p* < 0.001; P2 vs. P3, *p* < 0.001) (Figure 2c). The increase in the number of neurospheres at PND 48 was significant between adjacent passages except between P2 and P3 (PC vs. P1, *p* = 0.0307; PC vs. P2, *p* < 0.001; PC vs. P3, *p* < 0.001; P1 vs. P2, *p* < 0.001; P1 vs. P3, *p* < 0.001; P2 vs. P3, *p* = 0927) (Figure 2d).

Additionally, the number of vital cells in cell cultures over all passages was examined (n = 6) (Figure 3). The number of vital cells increased significantly between PC and P3 between all examined age groups (Figure 3a–d). At PND 8, PND 12, and PND 48, a significant increase was detected between all passages (PND 8: PC vs. P1, *p* < 0.001; PC vs. P2, *p* < 0.001; PC vs. P3, *p* < 0.001; P1 vs. P2, *p* < 0.001; P1 vs. P3, *p* < 0.001; P2 vs. P3, *p* < 0.001) (PND 12: PC vs. P1, *p* < 0.001; PC vs. P2, *p* < 0.001; PC vs. P3, *p* < 0.001; P1 vs. P2, *p* = 0.0060; P1 vs. P3, *p* < 0.001; P2 vs. P3, *p* = 0.0152) (PND 48: PC vs. P1, *p* = 0.0288; PC vs. P2, *p* < 0.001; PC vs. P3, *p* < 0.001; P1 vs. P2, *p* < 0.001; P1 vs. P3, *p* < 0.001; P2 vs. P3, *p* < 0.001) (Figure 3a,b,d). Changes between PC and P3 (*p* < 0.001), and P1 and P3 (*p* < 0.001), as well as P2 and P3 (*p* < 0.001) showed significance at PND 16 (Figure 3c). No significant change was detected between PC and P1 (*p* = 0.9898), PC and P2 (*p* = 0.7832), or P1 and P2 (*p* = 0.9190) at PND 16 (Figure 3c).

The diameter of the neurospheres was analyzed in all examined age groups and passages (Figure 4). In PC the diameter of the neurospheres was significantly larger at PND 8 (321.8 ± 10.65) and PND 16 (263.9 ± 10.65) (*p* < 0.001), and PND 8 and PND 48 (239.8 ± 6.98) (*p* < 0.001), as well as PND 12 (289.1 ± 11.20) and PND 48 (*p* = 0.0039). At PND 8 the diameter of the neurospheres significantly increased between P1 and P2 (*p* < 0.001) (Figure 4a). At PND 12, PND 16, and PND 48 the diameter of the neurospheres increased significantly between PC and P3 (P12, *p* = 0.0079; PND 16, *p* < 0.001; PND 48, *p* < 0.001) (Figure 4b–d). The diameters of the neurospheres were significantly larger between PC and P1 (*p* < 0.001), as well as PC and P2 (*p* < 0.001) at PND 12 (Figure 4b). At PND 16, a significant increase in the diameter of the neurospheres was observed between PC and P2 (*p* = 0.0185) (Figure 4c). The neurosphere diameter increased significantly between PC and P2 (*p* < 0.001), as well as P1 and P3 (*p* < 0.001) at PND 48 (Figure 4d).

The neurospheres were examined for the neural stem and progenitor cell markers nestin, doublecortin (DCX), Sox2, and Atoh-1 (Figure 5). The cytoplasm of cells within the neurospheres, as well as of emigrating cells, showed positive staining for the neural progenitor cell marker nestin at all investigated age groups of the MGB (Figure 5a–l). The transcription factor Atoh-1 was detected in colocalization with DAPI in the nuclei of cells from the neurospheres in PND 8, PND 12, PND 16, and PND 48 (5a–d). Similarly, the nuclei of the neurospheres from cells of the MGB showed positivity for the transcription factor and neural stem cell marker Sox2 at all age groups examined (Figure 5e–h). The neuronal migration marker DCX was detected in the cytoplasm of cells of the neurospheres of the MGB at PND 8, PND 12, PND 16, and PND 48 (Figure 5i–l).

### 2.2. The Differentiation Capacity of the MGB NSCs Reveals Age-Specific Characteristics

After dissociation of the neurospheres, the single cells obtained were stimulated to differentiate into glial (astrocytes and oligodendrocytes) and neuronal cells. These single cells were stained with specific markers to analyze their differentiation capacity (Figure 6). The single cells of the MGB were positive for ß-III-tubulin, which is a marker for a proneuronal fate of the cell, at all ages examined (Figure 6a). Additionally, astrocytes were detected in all age groups of the MGB using the glial marker GFAP (Figure 6b). Myelin basic protein (MBP) visualizes the myelination processes of oligodendrocytes. Single cells of the MGB showed positive staining for MBP at PND 8, PND 12, PND 16, and PND 48 (Figure 6c). As already observed in cells of the MGB neurospheres, the differentiated single cells showed positive staining for the neural progenitor cell marker nestin at all age groups examined (Figure 6d).

A quantitative evaluation of the differentiation marker positive single cells in relation to ß-tubulin positive cells was performed (Figure 7). The coverslips of four different cell cultures per age group were each analyzed by three examiners who are highly experienced in the evaluation of the differentiation markers. The entire glass coverslip, which was divided into 36 identical fields, was evaluated. The percentage of ß-III-tubulin positive cells was higher than 70% at PND 8, PND 12, and PND 48 (PND 8, 72.51 ± 4.096%; PND 12, 74.89 ± 5.567%; PND 48, 79.71 ± 2.162%) (Figure 7a). In contrast, at PND 16, the percentage of ß-III-tubulin positive cells was 6.743 ± 1.574% (Figure 7a). Quantification of the relative capacity to differentiate into astrocytes demonstrated increased astrocytic differentiation at PND 8 (7.653 ± 0.4704%) and PND 16 (10.68 ± 0.9648%). In contrast, a low percentage of astrocytic differentiation was present at PND 12 (0.356 ± 0.0885%) and PND 48 (0.7350 ± 0.1468%) (Figure 7b). At PND 8, the percentage of MBP-positive cells was 8.888 ± 1.783% (Figure 7c). Subsequently, a lower percentage of MBP-positive cells was found at PND 12 (2.295 ± 0.4626%), PND 16 (5.348 ± 0.9076%), and PND 48 (5.080 ± 0.8106%) (Figure 7c). The percentual evaluation of the progenitor cell marker nestin revealed an increase of approximately nine percentage points between PND 8 (18.40 ± 1.964%) and PND 12 (27.87 ± 1.693%), followed by a decrease to 4.493 ± 0.8153% at PND 16 (Figure 7d). At PND 48, the percentage of nestin was found to be 22.16 ± 1.318% (Figure 7d).

### 2.3. mRNA Abundance of Neurogenic Factors with High Relevance for MGB Neurogenesis Displays Age-Specific Patterns

Volcano plots were created to identify genes whose mRNA abundance provides both significant (*p* < 0.05) and relevant (fold change < 0.5 or >2) changes over time and to visualize the patterns of mRNA abundance in age comparison (Figure 8, Figure 9 and Figure 10). Comparing all age groups, the mRNA abundance of eight genes showed a significant and relevant increase with increasing age (Figure 8, Figure 9 and Figure 10). In contrast, the mRNA abundance of 20 genes showed a significant and relevant decrease in gene expression with increasing age (Figure 8, Figure 9 and Figure 10).

The comparison between the two youngest (PND 8 vs. PND 12) and between the youngest (PND 8) and oldest (PND 48) age groups was performed to highlight the extent of the differences in mRNA over time from early postnatal up to the adult stage (Figure 8). No neurogenic factor was detected between PND 8 and PND 12 whose difference in mRNA abundance was both relevant and significant (Figure 8a). The comparison between PND 8 and PND 48 showed a significant and relevant increase in the mRNA abundance of *interleukin 3* (*Il3*), *S100 calcium-binding protein B (S100b*), *hairy/enhancer-of-split related with YRPW motif-like (HeyL*), *leukemia inhibitory factor (Lif*), and *sonic hedgehog (Shh*) (Figure 8b). Furthermore, a significant and relevant decrease in the mRNA abundance of *doublecortin (Dcx*), *ephrin B1 (Efnb1*), *lactate dehydrogenase* (*Ldha*), *filamin A (Flna*), *microtubule-associated protein 2 (Map2*), *achaete-scute complex homolog 1 (Ascl1*), *neuropilin 1 (Nrp1*), *bone morphogenetic protein 2 (Bmp2*), *discs large homolog 4 (Dlg4*), *dopamine receptor D2 (Drd2*), and *histone deacetylase 4 (Hdac4*) was detected (Figure 8b).

In summary, the changes in neurogenetic factors at the mRNA level are very small between PND 8 and PND 12 (Figure 8a). In contrast, proastrocytic factors (*S100b* and *Lif*), but also NSC niche regulating factors (*Shh* and *HeyL*) show an increase in mRNA abundance with increasing age (Figure 8b). Proneural (*Efnb1*, *Map2*, *Ascl1*, *Bmp2*, *Drd2*, and *Hdac4*), migratory (*DCX*), and axogenetic (*Nrp1* and *Dlg4*) factors, on the other hand, display a significant decrease with increasing age (Figure 8b).

The comparison between PND 8 and PND 16 as well as PND 12 and PND 16 was made to highlight the differences in mRNA abundance around the hearing onset (Figure 9). Significantly changed and relevantly lower mRNA abundance at PND 16 than at PND 8 was found in *Dcx*, *neurotrophin 3 (Ntf3*), and *Efnb1* (Figure 9a). In contrast, *S100b*, *neurogenin 1 (Neurog1*), *fibroblast growth factor 2 (Fgf2*), *HeyL*, and *apolipoprotein E (Apoe*) had significantly altered. There was relevantly higher mRNA abundance at PND 16 than at PND 8 (Figure 9a). The comparison of the age groups PND 12 and PND 16 showed significantly altered and relevantly lower mRNA abundance of *glial cell-derived neurotrophic factor* (*Gdnf*), *Ntf3*, *Bmp2*, and *actin beta (Actb*) (Figure 9b). The mRNA abundance of *S100b* was significantly and relevantly higher at PND 16 than at PND 12 (Figure 9b).

The analysis of mRNA levels of neurogenetic factors around the hearing onset shows characteristic changes. There is a significant decrease in proneural (*Ntf3*, *Efnb1*, *Gdnf*, and *Bmp2*) and migratory factors (*Dcx*) after hearing onset (Figure 9a,b). In contrast, the mRNA of proastrocytic factors (*S100b* and *Apoe*) and factors that regulate the stem cell niche (*HeyL*, *FGF2*) are increased after hearing onset (Figure 9a,b).

A comparison of the age groups PND 12 and PND 16 with PND 48 showed a significant and relevant decrease in the mRNA abundance of multiple neurogenic factors at PND 48 (Figure 10). At PND 48, the mRNA abundance of *S100b* (PND 12 vs. PND 48) and *Il3* (PND 16 vs. PND 48) was significantly and relevantly increased (Figure 10a,b). The comparison of the age groups revealed a significantly and relevantly lower mRNA abundance at PND 48 than at PND 12 for the genes *pleiotrophin (Ptn*), *brain-derived neurotrophic factor (Bdnf*), *roundabout homolog 1 (Robo1*), *Efnb1*, *Ldha*, *Flna*, *Map2*, *Ascl1*, *Nrp1*, *Bmp2*, *Dlg4*, *Drd2*, *Actb*, *disheveled dsh homolog 3 (Dvl3*), *Hdac4*, and *notch homolog 1* (*Notch1*) (Figure 10a). The analysis of the age group comparison between PND 16 and PND 48 revealed an age-dependent low mRNA abundance of neurogenic factors (Figure 10b). The mRNA abundance of the genes *Ldha*, *Ascl1*, *Nrp1*, *Drd2*, *Dvl3*, *Hdac4*, *Notch1*, and *tyrosin hydroxylase (Th*) were significantly and relevantly decreased at PND 48 compared to PND 16 (Figure 10b).

With increasing age, an increasing number of neurogenetic factors show a decrease in mRNA abundance. Only the proastrocytic factor *S100b* and *Il3*, which influence neuritogenesis, show an increase in mRNA abundance (Figure 10a,b). The majority of neurogenetic factors whose mRNA abundance decreases significantly are proneural factors (*Ptn*, *Bdnf*, *Efnb1*, *Flna*, *Map2*, *Ascl1*, *Bmp2*, *Drd2*, *Dvl3*, *Hdac4*, *Th*, *Drd2*, and *Notch1*) (Figure 10a,b). Furthermore, the mRNA abundance of axogenetic factors (*Robo1*, *Nrp1* and *Dlg4*) decreases significantly with increasing age (Figure 10a,b).

## 3. Discussion

This study examined the rat MGB of several age groups for NSC capacity up to the adult stage. Indirect detection methods of the cardinal features of NSCs were performed because direct markers for NSCs do not exist [10,26]. The age groups represent crucial stages in the maturation of the rat and its auditory pathway. The hearing onset of the rat occurs between PND 10 and 14 [27,28,29]. The hearing onset had a decisive influence on the neurogenic potential of other nuclei of the auditory pathway [15,17]. Therefore, age groups before (PND 8), at (PND 12), and shortly after (PND 16) were chosen for the hearing onset. In addition, the MGB of rats older than 40 days was studied because domesticated rats reach sexual maturity at this time and are therefore referred to as adults [30]. Positive auditory brainstem responses of all vertex electrodes were recorded at PND 36, indicating a complete development of the auditory system of the rat [29]. Due to the ethical need to reduce the number of animals used in experiments, no further age groups were chosen.

### 3.1. The Maturing MGB Harbors a NSC Niche up to the Adult Stage with Decreasing Neurogenic Potential with Increasing Age

Under the influence of the NSC medium, which contains EGF and bFGF and is well established in NSC research, primary neurospheres formed in the cell cultures in all age groups examined (Figure 1) [10]. The number of neurospheres increased with increasing passage in all age groups and the number of neurospheres in P3 was significantly higher than in the PC (Figure 2). The evaluation of the vital cells in the neurosphere cultures likewise revealed a significant increase between PC and P3 at all ages (Figure 3). The results of the evaluation of the number of neurospheres and the number of vital cells in these cultures show a close correlation (Figure 3 and Figure 4). The diameter of the neurospheres is significantly larger in PND 8 and PND 12 compared to PND 48. This underlines the results from the analysis of the neurospheres and vital cells, as there is greater mitotic activity in the younger age stages. To investigate whether the decreasing neurosphere diameters are associated with slowly dying neurospheres over time, the diameters of the neurospheres were measured across all passages. This shows a significant increase between PC and P3 at all ages (Figure 4). An increase in the diameter of the neurospheres is known from studies of adulthood neurospheres of the dorsal vagal complex [31]. In summary, the capability of theoretically immortal mitotic self-renewal is present up to the adult stage in the cells of the MGB.

Subsequently, the neurogenic stem cell potential of the MGB is compared with the neurogenic stem cell potential of other adult NSC niches over time. For this purpose, the cardinal characteristics of NSCs are compared in detail. In the adult rat MGB, 197 neurospheres were formed per 100,000 cells (Figure 1). Few comparable studies have analyzed the number of neurospheres per 100,000 cells to assess the proliferative capacity of the NSC niche. In the dorsal vagal complex of the adult rat, 760 neurospheres per 100,000 cells were formed [31]. In the auditory pathway, 68 neurospheres per 100,000 cells were reported in the inner ear of the adult mouse [32]. The adult rat cochlear nucleus demonstrated the formation of 1 neurosphere per 100,000 cells [17]. In the adult inferior colliculus of the rat, 348 neurospheres per 100,000 cells were formed [15]. Compared to the proliferative capacity of the NSC niches of other adult auditory nuclei, the proliferative capacity of the NSCs of the adult MGB is prominent. Interestingly, the number of neurospheres, indicating the proliferation capacity, decreases significantly and pronouncedly after PND 12, corresponding to the hearing onset (Figure 1). This represents a crucial difference compared to the proliferative ability of the NSCs of other auditory nuclei. In the cochlear nucleus, located in the brain stem, and in the inferior colliculus, situated in the midbrain, there is a significant decrease in proliferative capacity even before hearing onset [15,17].

### 3.2. The Capacities to Form Progenitor Cells and to Differentiate into Neurons and Glial Cells of MGB NSCs Differs with Increasing Age

Further cardinal features of NSCs are the forming of progenitor cells and their differentiation into neurons and glial cells [10]. The neural progenitor marker nestin is an intermediate filament expressed from the early stages of neuroepithelial cell development to the terminal differentiation of the cells [33]. In addition, the NSC markers Atoh-1, Sox2, and DCX were detected in the neurospheres of the MGB at all ages examined (Figure 5). The basic helix–loop–helix (bHLH) transcription factor Atoh-1 fulfills an essential function during the development of the auditory system [34,35]. Sox2 is detected in NSCs at all developmental stages and is a transcription factor of the high mobility group (HMG) box [36]. The neuronal migration protein DCX is a microtubule-associated protein expressed by differentiating and migrating precursor cells [37].

Comparing the relative expression of the neural progenitor marker nestin, the drop in expression at PND 16 (4.493 ± 0.8153%) is evident (Figure 7d). At PND 12 and PND 48, the percentage of nestin-positive cells is 27.87 ± 1.693% and 22.16 ± 1.318% (Figure 7d). A potential explanation for the decrease in the proportion of progenitor cells at PND 16 is that, compared to other auditory nuclei, the proliferation capacity of the MGB NSCs is pronounced up to the hearing onset (Figure 1). This is an indication that auditory–sensory signals stimulate NSCs to differentiate and influence the size of the stem cell pool. Interestingly, it has already been shown that the sensory influence on neural stem cell niches has different effects. Chemosensory niches such as those of the olfactory bulb receive survival signals through sensory stimulation, whereas the size of the stem cell niche in stem cell niches of the visual system is influenced by sensory stimuli [38]. It is known that the auditory and visual systems share important characteristics and that sensory input stimulates similar changes in both systems [39]. The percentage of nestin in histologic sections of the mouse auditory cortex was about 30% at PND 14 and decreased at later stages [16]. The relative proportion of nestin-positive cells in the NSCs of the inferior colliculus was less than 15% in comparable age groups [15]. Thus, the potential of the MGB NSCs to form neural progenitor cells over time lies between the potential of the inferior colliculus and the auditory cortex, consistent with its anatomical location.

The dissociated cells from neurospheres of the MGB expressed neuronal markers and glial markers at PND 8, PND 12, PND 16, and PND 48 under the influence of the differentiation medium (Dif medium) (Figure 6a–c). The microtubule protein ß-III-tubulin, which is almost exclusively expressed by neurons in the early developmental stages, indicates the proneural fate of the cells [40]. GFAP, which belongs to the intermediate filaments, is expressed by astrocytes [41]. The myelination processes of oligodendrocytes are represented by MBP [42]. A comparison of the capacity of MGB NSCs to differentiate into neurons and glial cells reveals similarities and differences compared to neighboring auditory nuclei. The proneural differentiation capacity of the inferior colliculus is lower before and at hearing onset (50.97% at PND 6 and 50.98% at PND 12) than at comparable ages of the MGB (Figure 7a) [15]. Interestingly, the relative proportion of ß-III-tubulin-positive cells decreases in the inferior colliculus after hearing onset, as in the MGB. However, the percentage of proneurally differentiated cells at the adult stage in the MGB is similar to that at hearing onset (Figure 7a). In the inferior colliculus, the percentage decreases further at the adult stage [15]. Likewise, it is possible that experimental causes influence the results of the ß-III-tubulin analysis at PND 16. Various factors influence the maturation and differentiation of cells in in vitro studies [43]. Cell culture studies influence the cell cycle and thus present a risk of selection bias. Cell–cell contacts, surface interactions, and growth factors ensure the survival of certain cell populations and the decline of other populations [44]. In addition, it should be considered that a development-dependent post-translational modification of ß-III-tubulin takes place and thus different isoforms of ß-III-tubulin are present at different ages [45]. The capacity for astrocytic differentiation is similar but reversed. In the MGB and inferior colliculus, there is an increase after the hearing onset (PND 16 and PND 24) (Figure 7b) [15]. However, in the MGB, the relative proportion of GFAP-positive cells decreases again in the adult stage. These results indicate that in the MGB, in contrast to the neighboring inferior colliculus, more proneuronal rather than astrocytic differentiation of NSCs occurs in the adult stage. Differences in the regulation of the NSC population and the neuronal differentiation capacity between NSC niches are known from the literature. This is highlighted in the two NSC niches intensively analyzed: the hippocampus and the subventricular zone. Hippocampal NSCs undergo asymmetric division, giving rise to new neurons at the expense of NSCs [46]. In contrast, simultaneous self-renewal and neuronal differentiation occur in NSCs of the subventricular zone [47]. The relative differentiation from oligodendrocytes, which MBP indicates, is comparable to the inferior colliculus [15]. An age-related decline in MBP is known from studies of the auditory nerve in mice [48]. It should be noted that NSCs are not a homogeneous cell population but a combination of diverse subpopulations. This makes it difficult to compare NSC niches with each other, as there are regional differences [49]. NSCs that exist in different areas of the brain have different functions and therefore express different markers [50,51]. This should be consistently considered when assessing proliferation and differentiation studies of NSCs.

The cardinal characteristics of NSCs were detected in cells of the MGB up to the adult stage. NSCs represent a potential target for regenerative therapy approaches in the auditory system. Electrical stimulation by a cochlear implant induces NSC-derived neural regeneration [52]. Patients with cochlear implants experience an improvement in hearing function many years after implantation [53]. One possible explanation for this is the existence of NSCs in the auditory nuclei, which ensure relevant neuroplasticity. Interestingly, translational approaches are increasingly being pursued to treat neuronal deficits with NSCs [54]. It has been shown that patients with cerebral infarction subsequently produce new neurons [55]. Advances in the field of neurosurgical intervention at the MGB give hope that a drug or stem-cell-based therapy might become available [24]. Insights into the neurogenesis and capacities of NSCs of the MGB represent a basis for this.

### 3.3. mRNA Levels of Prominent Neurogenic Factors Reflect the NSC Potential of the MGB

Additionally, molecular genetic studies were performed to analyze the mRNA levels of neurogenetic factors and their age-dependent dynamics in the native tissue of the MGB. The mRNA abundance of 84 genes with crucial relevance for neurogenesis and NSCs in the rat MGB was analyzed over time up to the adult stage. The individual comparison of the age groups enables an analysis of the influence of the developmental stages of the auditory system on MGB neurogenesis and the identification of single neurogenic factors with potential impact on the development of the rat MGB.

First, the mRNA abundance of the age groups adjacent to the hearing onset is compared to highlight the crucial importance of the hearing onset for the maturation of the auditory pathway. When comparing PND 8 and PND 12, there was no significant and relevant difference among the mRNA abundance of the 84 neurogenic factors (Figure 8a). The volcano plots of PND 12 and PND 16 demonstrate a significant and relevant increase of the proastrocytic factor *S100b* and a significant and relevant decrease of *Ntf3*, which regulates progenitor cell activation, of *Gdnf*, which ensures neuron survival, and of *BMP2*, which provides proneural influence (Figure 9b) [56,57,58,59]. The significant and relevant differences in mRNA abundance at PND 8 and PND 16 provide additional information about the changes in mRNA after hearing onset (Figure 9a). Overlapping with the results of the evaluation of PND 12 vs. PND 16, the progenitor cell activating factor *Ntf3* showed a decrease and the proastrocytic differentiation factor *S100b* showed an increase in mRNA abundance at PND 8 vs. PND 16 (Figure 9a). The neuronal migration and proliferation factors *Dcx* and *Efnb1* reveal a significant and relevant decrease in mRNA abundance at PND 16 compared to PND 8 (Figure 9a) [37,60,61]. *Neurog1* showed a significant and relevant increase in mRNA at PND 16 compared to PND 8 and plays an important role in the neuronal maturation of the auditory pathway (Figure 9a) [62]. An increase in *Neurog1* mRNA abundance is unexpected because a decrease in neuronal differentiation has been observed at the protein level and *Neurog1* is known as a proneural factor (Figure 7a) [63]. Interestingly, these results are consistent with current research and suggest *Neurog1* is a negative regulator of neuronal differentiation [64]. The mRNA abundance of *Apoe* is higher at PND 16 than at PND 8 (Figure 9a). *Apoe* is classically expressed by astrocytes [65]. *HeyL* is part of the *bHLH* transcription factor family and promotes the proneuronal differentiation of neuronal progenitor cells [66]. *Hey* genes, among others, are highly expressed in the maturing dorsal root ganglion, where they regulate neuronal differentiation and are subject to complex internal interactions [67,68,69]. The increase in *HeyL* mRNA abundance after hearing onset indicates that *HeyL* plays an important role in the neuronal differentiation of MGB NSCs. A significant and relevant increase in the mRNA abundance of *Fgf2*, which suppresses the proastrocytic differentiation of NSCs and maintains NSC potential, was observed in PND 8 vs. PND 16 (Figure 9a) [70]. *Fgf2* overexpression results in an increased pool of immature and migrating neurons suitable for neuronal repair [71]. The endogenous increase in *Fgf2* mRNA abundance after the NSC niche stimulating phase at the hearing onset (PND 16) suggests an endogenous regulation at the mRNA level.

In addition to the changes in NSC capacity around the hearing onset, the persistence of the NSC niche into the adult stage is a phase of great interest. A comparison of the mRNA abundance of neurogenic factors between the age groups PND 8, 12, and 16, and PND 48 of the MGB was performed.

The mRNA abundance of the proastrocytic differentiation factor *S100b* (PND 8 vs. PND 48, PND 12 vs. PND 48) and *Lif* (PND 8 vs. PND 48) increased significantly and relevantly with age (Figure 8b and Figure 10a) [56,72]. The mRNA abundance of *Il3* (PND 8 vs. PND 48, PND 16 vs. PND 48) increases significantly and relevantly with increasing age (Figure 8b and Figure 10b). *Il3* provides for increased neuritogenesis in the central nervous system [73]. These data indicate that *Il3* plays a vital role in the adult neuritogenesis of the MGB. In addition, the neuronal proliferation, differentiation, and migration factors *Dcx* (PND 8 vs. PND 48), *Efnb1* (PND 8 vs. PND 48, PND 12 vs. PND 48), and *Bdnf* (PND 12 vs. PND 48) decrease significantly and relevantly with increasing age (Figure 8b and Figure 10a) [37,60,74]. Accordingly, the factors *Nrp1* (PND 8 vs. PND 48, PND 12 vs. PND 48, PND 16 vs. PND 48), *Dlg4* (PND 8 vs. PND 48, PND 12 vs. PND 48), and *Robo1* (PND 12 vs. PND 48), which are involved in axogenesis and synaptogenesis, show significantly and relevantly reduced mRNA abundance at PND 48 in comparison with the younger age groups (Figure 8b and Figure 10a,b) [75,76,77]. The neurogenic factors *Map2* (PND 8 vs. PND 48, PND 12 vs. PND 48), *Bmp2* (PND 8 vs. PND 48, PND 12 vs. PND 48), and *Th* (PND 16 vs. PND 48) are important in neuronal development and proneuronal differentiation, and display significantly and relevantly decreasing mRNA abundance with increasing age (Figure 8b and Figure 10a,b) [58,78,79]. Furthermore, a significant and relevant decrease in the mRNA abundance of factors that promote neuronal survival was demonstrated. The neurogenic and neuronal survival-promoting factors *Hdac4* (PND 8 vs. PND 48, PND 12 vs. PND 48, PND 16 vs. PND 48) and *Ptn* (PND 12 vs. PND 48) had significant and relevant lower mRNA abundance at PND 48 than at the younger age groups (Figure 8b and Figure 10a,b) [80,81]. The mRNA abundance of the neurogenic factors *Dvl3* (PND 12 vs. PND 48, PND 16 vs. PND 48), which has essential functions in cochlear development and neurogenesis, and *Actb* (PND 12 vs. PND 16, PND 12 vs. PND 48), which is responsible for the neuronal cytoskeleton and is known to decrease with age, were significantly and relevantly reduced with increasing age [82,83,84]. *Ldha*, which shows a significant and relevant decrease in mRNA abundance (PND 8 vs. PND 48, PND 12 vs. PND 48, PND 16 vs. PND 48) with increasing age, plays an essential role in aging processes in the rodent brain [85]. An increase in proastrocytic factors and a decrease in proneuronal neurogenic factors at the mRNA level of the MGB was detected with increasing age at the adult stage (PND 48).

In a comparison of PND 8 and PND 48, the mRNA abundance of *HeyL*, which promotes the proneuronal fate of NSCs, and the mRNA abundance of *Shh*, which has a decisive influence on the maintenance of the NSC niche, increase (Figure 8b) [86,87,88]. *Shh* is a neurogenic factor that regulates the number of cells with the NSC character in the NSC niche [89]. In contrast, the mRNA abundance of *Drd2*, which has an essential influence on the regulation of neural progenitor cells, decreases significantly and relevantly (PND 8 vs. PND 48, PND 12 vs. PND 48, PND 16 vs. PND 48) (Figure 8b and Figure 10a,b) [90,91]. Similarly, the mRNA abundance of *Flna* (PND 8 vs. PND 48, PND 12 vs. PND 48), *Ascl1* (PND 8 vs. PND 48, PND 12 vs. PND 48, PND 16 vs. PND 48), and *Notch1* (PND 12 vs. PND 48, PND 16 vs. PND 48) decreases significantly and relevantly with increasing age (Figure 8b and Figure 10a,b). These factors have both a stimulating and a maintaining influence on the NSC niche [92,93,94].

The amount of the mRNA abundance of several neurogenic factors decreases significantly with increasing age. It is reasonable to assume that, on the mRNA level, the MGB harbors a decreasing neurogenic potential of the MGB with increasing age, which confirms the in vitro results. It is known that the adult NSC niche is regulated in a complex manner. In addition to intrinsic factors such as transcription factors and cytoplasmic factors, extrinsic regulators of the stem cell niche exist [95,96]. Likewise, the epigenetic regulation of the adult NSC niche plays a vital role [97]. Adult NSC niches are subject to significant heterogeneity and are regulated in complex and different ways [51]. Despite the complex interplay of neurogenetic factors, individual factors were identified whose differences in mRNA abundance over time were significant and relevant. Recent developed technologies in stem cell research provide promising results. Over-expression of key genes helped to potentiate the regenerative potential of NSCs. Both growth factors such as BDNF and transcription factors such as Nurr1 were shown to be effective in functional regeneration after genome editing [98,99]. Understanding of the neurogenetic factors that influence the physiological development of MGB NSCs represents an interesting basis for therapeutic approaches of this kind.

### 3.4. Limitations of the Study

This study has several limitations. The comparison of the proliferation capacity, investigation of the neurospheres, and passage analysis of different studies is limited. The studies differ in the animal model, the cell culture medium used, the time points and passaging technique, and the evaluation method. These factors have a relevant influence on the number of neurospheres. There is evidence that cell culture studies of neurospheres, especially under the influence of growth factors, promote the selection advantages of individual cell subpopulations [43]. Cell–cell interactions and the phase of the cell cycle which individual cells are in play a crucial role [43]. Likewise, there is a risk that the survival preferences of individual cell subpopulations effected by different surface chemistry influence the evaluation in NSC studies [44]. The results of the differentiation examinations at PND 16 were repeated, as they differed noticeably from the findings of the other age groups, particularly in proneuronal and proastrocytic differentiation. The results of the two measurements showed close correlations and their mean values are shown (Figure 7). ß-III-tubulin is subject to a pronounced post-translational modification up to PND 30 and therefore it is possible that other proneural markers are increasingly expressed at PND 16. Similarly, no specific neurons were generated in this study that take on specific tasks in the auditory system. This study should form the basis for further studies in which specific neurons are generated and their functionality in the auditory system is analyzed [45]. This study provides strong evidence for an adult MGB NSC niche. Whether this NSC niche is active or quiescent has yet to be answered and is subject to further investigations. This study should serve as a basis for further studies investigating the MGB NSC niche and its molecular genetic regulation.

## 4. Materials and Methods

### 4.1. Animal Preparation, Neurosphere Assay, and Passaging

Postnatal day (PND) 8, 12, 16, and adult (PND 48) Sprague–Dawley rats (Charles River^®^, Wilmington, MA, USA) were euthanized by cervical dislocation. MGB was microscopically (OPMI1, Zeiss^®^, Oberkochen, Germany) dissected and transferred into Neurobasal^®^ medium (Thermo-Fisher Scientific^®^, Grand Island, NE, USA) at room temperature. The preparation steps were followed according to the protocol described previously [20]. All procedures were performed under antiseptic conditions. Equal numbers of females and males were chosen for each age group.

All experimental procedures were performed in accordance with the ethical guidelines for the use of animals in experiments of the European Committees Council Directive (2010/63/EU) and were approved by the local animal care committee of the Government of Lower Franconia.

Following preparation, MGB was transferred to Accutase (Gibco^®^, Thermo Fischer Scientific^®^, Grand Island, NE, USA). Tissue was dissociated enzymatically in a ThermoMixer^®^ (Eppendorf^®^, Hamburg, Germany) at 37 °C and 500 rpm with trituration steps every 10 min. Trituration with a 500 μL pipette was performed until an emulsion formed and no macroscopic tissue parts were visible. After dissociation, cells were centrifuged at 1000 rpm for 5 min (Centrifuge 5810, Eppendorf^®^, Hamburg, Germany). The resulting pellet was resuspended in neural stem cell medium (NSC medium) containing serum-free Neurobasal^®^ medium (Thermo-Fisher Scientific^®^, Grand Island, NE, USA), 1% GlutaMAX^®^ (Gibco^®^, Thermo Fischer Scientific^®^, Grand Island, NE, USA) supplement, 2% B27^®^ supplement without retinoic acid (Gibco^®^, Thermo Fischer Scientific^®^, Grand Island, NE, USA), 1% penicillin/streptomycin (Gibco^®^, Thermo Fischer Scientific^®^, Grand Island, NE, USA) and 10 ng/mL recombinant murine growth factors EGF and bFGF/FGF-2 (PeproTech^®^, Thermo Fischer Scientific^®^, Grand Island, NE, USA). For some of the studies, 100,000 cells per cell culture were used. However, the passage studies added the number of cells to the cell cultures. Cell count was determined on 10 μL of the single cell suspension mixed with 10 μL of trypan blue (Thermo Fischer Scientific^®^, Grand Island, NE, USA). Thus, the number of vital cells was counted in an improved Neubauer hemocytometer (ZK06, Hartenstein^®^, Wuerzburg, Germany) and 100,000 cells per animal were added to the cell culture. Cell suspension with 4 mL NSC medium was transferred into hydrophobic 50 mL/25 cm^2^ filter top cell culture flasks (CELLLSTAR^®^, filter top, 25 cm^2^, Greiner^®^ Bio-One, Monroe, NC, USA) and cultivated as free-floating cell cultures at 37 °C and 5% CO_2_. Fresh NSC medium (2 mL) was added every four days. The number of neurospheres was determined before passaging and after 30 days in culture with an inverted microscope (Leica^®^ DMI 4000B and DMI-8, Wetzlar, Germany) at 5× magnification. Neurospheres were mechanically dissociated for passage and the single-cell suspensions were centrifuged for 5 min. at 1000 rpm (Centrifuge 5810, Eppendorf^®^, Hamburg, Germany). The cell pellet was rinsed once with PBS solution (Gibco^®^, Thermo Fischer Scientific^®^, Grand Island, NE, USA), resuspended in fresh NSC medium, and cultivated as described above. The absolute number of vital cells, the number of neurospheres, and the neurosphere diameters were determined before each passaging step, as mentioned above.

### 4.2. Plating of Neurospheres and Differentiated Single Cells

Neurospheres from the free-floating cell cultures were carefully absorbed with a 5 mL automatic pipette (Multipette plus, Eppendorf^®^, Hamburg, Germany) and plated onto glass coverslips (78.5 mm^2^, Hartenstein^®^, Wuerzburg, Germany) coated with Poly-D-lysine (100 μg/mL, Serva Electrophoresis^®^, Thermo Fischer Scientific^®^, Grand Island, NE, USA) and Laminin-1 (10 μg/mL, BD Biosciences^®^, Bergen County, NJ, USA). Plated neurospheres were cultivated in 4-well dishes (Greiner^®^ Bio-One^®^, Monroe, NC, USA) with 100 μL of NSC medium per coverslip for 24 h at 37 °C/5% CO_2_.

For the plating of single cells, neurospheres were dissociated mechanically. Complete dissociation of the neurospheres into single cells was checked using an inverted transmitted light microscope (Leica^®^, DMI-8, Wetzlar, Germany). The single-cell suspension was centrifuged at 1000 rpm for 5 min. The cell pellet was rinsed once with PBS solution (Gibco^®^, Thermo Fischer Scientific^®^, Grand Island, NE, USA) and after removal of PBS solution NSC medium was added. The single cells were plated on glass coverslips (78.5 mm^2^, Hartenstein^®^, Wuerzburg, Germany) and coated first with Poly-d-lysine (100 μg/mL, Serva Electrophoresis^®^, Thermo Fischer Scientific^®^, Grand Island, NE, USA) for 1 h at room temperature and afterwards with Laminin-1 (10 μg/mL, BD Biosciences^®^, Bergen County, NJ, USA) at 37 °C/5% CO_2_ for 1 h at a density of 100 cells/mm^2^ (8000 cells/coverslip) and cultivated in 4-well dishes (Greiner^®^ Bio-One^®^, Monroe, NC, USA) for 3 days. After 3 days NSC medium was removed and a differentiation medium (DIF medium) was added, consisting of Neurobasal^®^ (Thermo-Fisher Scientific^®^, Grand Island, NE, USA), 1% GlutaMAX^®^ (Gibco^®^, Thermo Fischer Scientific^®^, Grand Island, NE, USA) supplement, and B27 with retinoic acid (Thermo-Fisher Scientific^®^, Grand Island, NE, USA). For differentiation experiments, the single cell suspension in DIF medium was for 7 days at 37 °C/5% CO_2_. The cell count was determined on a 10 μL sample, admixed with 10 μL of trypan blue (Thermo Fischer Scientific^®^, Grand Island, NE, USA) as described in 4.1. DIF medium was changed completely every 2 days.

### 4.3. Fixation and Immunocytochemistry

Neurospheres and cells on glass coverslips were fixed after the experiment with a 4% paraformaldehyde solution (PFA in 0.1 M NaPP; Sigma-Aldrich^®^, Merck^®^, St. Louis, MO, USA) for 30 min. Non-specific binding sites were blocked by a solution of 10% bovine serum albumin (BSA, A9418 Sigma-Aldrich^®^, Merck^®^, St. Louis, MO, USA) in 0.1 M PBS buffer solution (Sigma-Aldrich^®^, Merck^®^, St. Louis, MO, USA). The following primary antibodies were used incubation at 5 °C for 12 h in 1% BSA solution and 0.1 M PBS buffer: mouse monoclonal against Atoh-1 (1:1000; Ab27667—Abcam^®^, Waltham, MA, USA), mouse monoclonal against β-tubulin (1:1000; #TS293—Sigma-Aldrich^®^, Merck^®^, St. Louis, MO, USA), mouse monoclonal against β-III-tubulin (1:1000; #Ab7751—Abcam^®^, Waltham, MA, USA), rabbit polyclonal against β-III-tubulin (1:2000; #Ab18207—Abcam^®^, Waltham, MA, USA), rabbit polyclonal against doublecortin (DCX) (1:1000; #Ab18723—Abcam^®^, Waltham, MA, USA), mouse monoclonal against the glial fibrillary acidic protein (GFAP) (1:1000; #MAB360—Merck Millipore^®^, Billerica, MA, USA), rabbit polyclonal against myelin basic protein (MBP) (1:800; #M3821—Sigma-Aldrich^®^, Merck^®^, St. Louis, MO, USA), mouse monoclonal against Nestin (1:800; #MAB353—Merck Millipore^®^, Billerica, MA, USA) and rabbit polyclonal against Sox2 (1:2000; #Ab97959—Abcam^®^, Waltham, MA, USA). Three washing steps were performed in 0.1 M PBS solution (Sigma-Aldrich^®^, Merck^®^, St. Louis, MO, USA). Afterwards, incubation with the second antibodies coupled to Alexa Fluor A488 or A555 (1:1000, #A11001, #A11008—Thermo-Fisher^®^, Grand Island, NE, USA) was performed with 5 μg/mL DAPI (1:5000, D9542, Sigma-Aldrich^®^, Merck^®^, St. Louis, MO, USA) for 1 h in a 1% solution of BSA (A9418 Sigma-Aldrich^®^, Merck^®^, St. Louis, MO, USA) and 0.1 M PBS (Sigma-Aldrich^®^, Merck^®^, St. Louis, MO, USA). Three final washing steps were performed in 0.1 M PBS solution (Sigma-Aldrich^®^, Merck^®^, St. Louis, MO, USA). The glass coverslips were embedded on object slides with Mowiol^®^ (4-88, Sigma-Aldrich^®^, Merck^®^, St. Louis, MO, USA) and stored at 5 °C in the dark.

### 4.4. Digital Images and Cytological Analysis

Digital images of the immunocytochemically stained neurospheres and differentiated single cells were taken with a Leica^®^ DMI-8 fluorescence microscope and Leica^®^ Application Suite X software v3.0.1 (Leica^®^, Wetzlar, Germany). Quantitative analysis of the immunocytochemically stained preparations on the glass coverslips was performed using a transmitted light technique with a 20× lens in tile-scan mode. The quantitative analysis of the immunocytochemically stained preparations on the glass coverslips was performed independently by 3 examiners who are highly experienced in the evaluation of the differentiation markers. The acquired image files were analyzed with Fiji/ImageJ V2.0.0 software [100]. The final images were composed using Adobe^®^ InDesign CC 2023 v18.1 software (Adobe Inc., San Jose, CA, USA).

### 4.5. RNA Extraction and cDNA Synthesis

After preparation, two paired MGBs per animal were transferred to DNA-, DNase-, RNase-, and pyrogen-free cryovials (Simport Scientific^®^, Saint-Mathieu-de-Beloeil, QC, Canada) and placed immediately in liquid nitrogen for at least 15 min. After 15 min, the tissue was transferred into bead-filled tubes (Precellys^®^ Lysing Kit CK 14, Bertin, France). MGBs of the same age group were pooled in one bead tube (n = 4 animals and 8 MGBs per age group). Pooling was performed to generate enough mRNA for further analysis. Regardless of age, the pooled MGBs weighed (Sartorius^®^ Handy M160, Goettingen, Germany) less than 20 mg. An amount of 350 µL of RLT buffer (Qiagen^®^, Venlo, The Netherlands) was added per bead tube. The tubes were homogenized in two homogenizer steps (Precellys 24 DUAL^®^, Bertin, France) at 6000 rpm for 30 s each. The resulting emulsion was mixed with 350 µL of ethanol 70% (Thermo Fisher Scientific^®^, Waltham, MA, USA). The following steps were performed according to the instructions of the RNeasy Mini Kit (Qiagen^®^, Venlo, The Netherlands). Subsequently, the extracted RNA was quantified using NanoDrop One/One^c^ spectrophotometer (Thermo Fisher Scientific^®^, Waltham, MA, USA) and its purity (A260/A280-Ratio) was determined. Animals had approximately 150 ng/mL RNA at p8, about 200 ng/mL RNA at p12, about 260 ng/mL RNA at p16, and about 10 ng/mL RNA at the adult stage. RNA with an A260/A280 ratio of 2.0 ± 0.2 was used to synthesize complementary DNA (cDNA).

Further steps were carried out according to the RT^2^ First Strand Kit (Qiagen^®^, Venlo, The Netherlands). An amount of 10 µL of DNA elimination mix was prepared with 500 ng RNA, Buffer GE, and RNase-free water per reaction, and incubated for 5 min at 42 °C (Biometra Trio 30, Analytik Jena^®^, Jena, Germany). After placing on ice for 1 min, 10µL reverse transcriptase mix with 2µL RE3 reverse transcriptase (Qiagen^®^, Venlo, The Netherlands) was prepared and added into the cooled DNA elimination mix. The mixture was incubated at 42 °C for 15 min and immediately followed by incubation at 95 °C for 5 min (Biometra Trio 30, Analytik Jena^®^, Jena, Germany). Finally, 91 µL RNase-free water was added per sample and the samples were stored at −20 °C for a maximum of eight weeks.

### 4.6. Rat Neurogenesis RT^2^ Profiler^TM^ PCR Array

A solution containing 102 µL from the cDNA obtained in the previous step, 1350 µL 2XRT2 SYBR Green Mastermix (Qiagen^®^, Venlo, The Netherlands), and 1248 µL RNase free water was produced. Per well, 25 µL of this suspension was transferred to the 96-well Rat Neurogenesis RT2 Profiler^TM^ PCR Array (PARN-404ZC-12, Qiagen^®^, Venlo, The Netherlands), which was sealed afterward. This array contains a primer set for 84 genes related to neurogenesis and NSCs. Additionally, the array comprises 5 reference genes (*actin beta*, *beta-2-microglobulin*, *hypoxanthine phosphoribosyltransferase 1*, *lactate dehydrogenase A*, and *ribosomal protein P1*), 3 reverse transcription controls (RTCs), 3 PCR reproducibility controls (PPCs) and 1 contamination control (GDC). The following steps were performed using the real-time PCR system StepOnePlus^TM^ (Thermo Fisher Scientific^®^, Waltham, MA, USA). Threshold values were identical for all analyses performed and the automated baseline option of the system was used as a baseline. The following cycling conditions were used: 10 min at 95 °C for denaturation, 40 cycles at 95 °C for 15 s, and 60 °C for 1 min. The cycle threshold (Ct) determined this way was exported as a spreadsheet calculation from Microsoft^®^ Excel 2023 V16.70 (Microsoft Corporation, Redmond, WA, USA). For each age group, 3 replicates were performed from the pooled samples.

### 4.7. Statistical Analysis

All collected data were compiled using Microsoft^®^ Excel 2023 V16.70 (Microsoft Corporation, Redmond, WA, USA) spreadsheets and statistically analyzed with GraphPad^®^ Prism 9.5.0 software (Graphpad Software Inc., San Diego, CA, USA). First, a column analysis (Shapiro–Wilk and Kolmogorov–Smirnov normality tests) was performed to determine whether a Gaussian normal distribution of the data was present. Subsequently, data were analyzed using the ordinary one-way ANOVA test followed by a Tukey multiple comparison test. A *p*-value < 0.05 was considered to be statistically significant. Reproducible results were obtained from six or more samples. If the data set followed a Gaussian normal distribution, the mean and standard error of the mean (SEM) are displayed. In contrast, the mean and standard deviation (SD) are depicted without a Gaussian normal distribution.

Ct values were analyzed using the Qiagen^®^ GeneGlobe Data Analysis Web Portal (Qiagen, Venlo, The Netherlands). All samples passed the PCR array reproducibility test (ΔCt average RTC—average PPC ≤ 5), the reverse transcription efficiency test (Ct PPC of the three replicates within an array is 20 ± 2 or the average PPC CT values of any 2 arrays do not differ by more than 2), and the genomic DNA contamination test (Ct GDC ≥ 35). Each gene was normalized with the arithmetic mean of the reference genes *beta-2-microglobulin* and *hypoxanthine phosphoribosyltransferase 1* to obtain the ΔCt value. The other reference genes showed significant or relevant changes between the different age groups and were therefore excluded as reference genes. ΔΔCt was calculated for each gene by subtracting the ΔCt value of the age group (n = 3) from the ΔCt value of the control age group (n = 3). The fold change for each gene from the age group to the control age group was calculated as 2^(−ΔΔCt)^. This procedure has been performed in a familiar and established manner [101]. *p*-values were calculated using a Student’s *t*-test based on the ΔCt values of the replicates for each gene in each age group compared to the control age group. The volcano plot allows the evaluation of relevant changes in mRNA abundance in the context of their statistical significance. It represents on the *x*-axis the log base 2 of the fold change value of each gene and on the *y*-axis the negative log base 10 of the *p*-value of the gene. The graphical representations of the volcano plots and the bar charts were created with GraphPad^®^ Prism 9.5.0 software (Graphpad Software Inc., San Diego, CA, USA). Colors were chosen according to the recommendations of current literature to provide the best accessibility for color-blind readers [102]. Data generated can be accessed under Appendix A (Appendix A: Gene Table, Appendix A: Average Ct, Appendix A: Average Delta(Ct), Appendix A: 2^(−Avg.(Delta(Ct)), Appendix A: Fold Change, Appendix A: *p*-value). The final images were composed using Adobe^®^ InDesign CC 2023 v 18.1 software (Adobe Inc., San Jose, CA, USA)).

## 5. Conclusions

In summary, all cardinal features of NSCs are detectable from PND 8 to PND 48 in the cells of the MGB and provide strong evidence for a persistent adult NSC niche in the MGB. The results presented in this study provide strong evidence that a neurogenic stem cell niche with a decreasing neurogenic potential persists in the MGB up to the adult stage.

Thus, the adult MGB NSC niche represents an interesting target for stimulating endogenous NSCs for proneural differentiation.

## Figures and Tables

**Figure 1 ijms-25-02623-f001:**
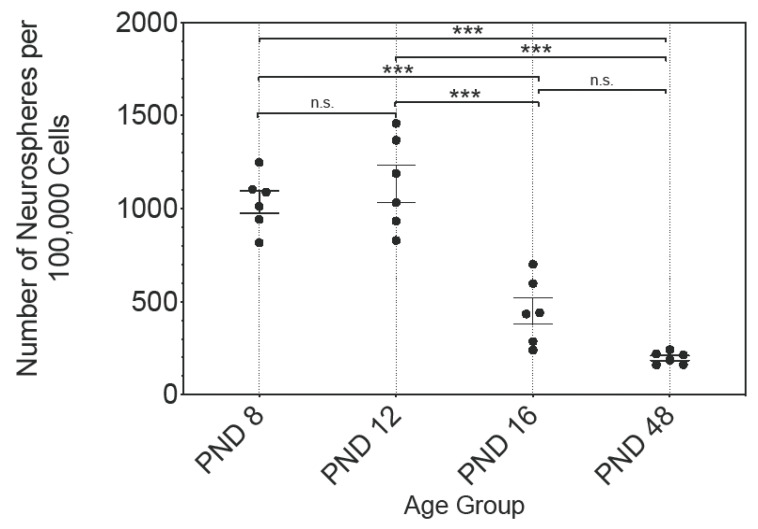
The analysis of the ratio of the number of primary neurospheres after 30 days in culture to 100,000 seeded cells of the rat MBG displays an age-dependent decrease. The most pronounced decrease in the number of neurospheres was between PND 12 and PND 16. The central horizontal bars show the mean; error bars depict the standard error of the mean (SEM); each dot represents a cell culture, n = 6; asterisks indicate the level of significance: n.s. = not significant, *** *p* < 0.001.

**Figure 2 ijms-25-02623-f002:**
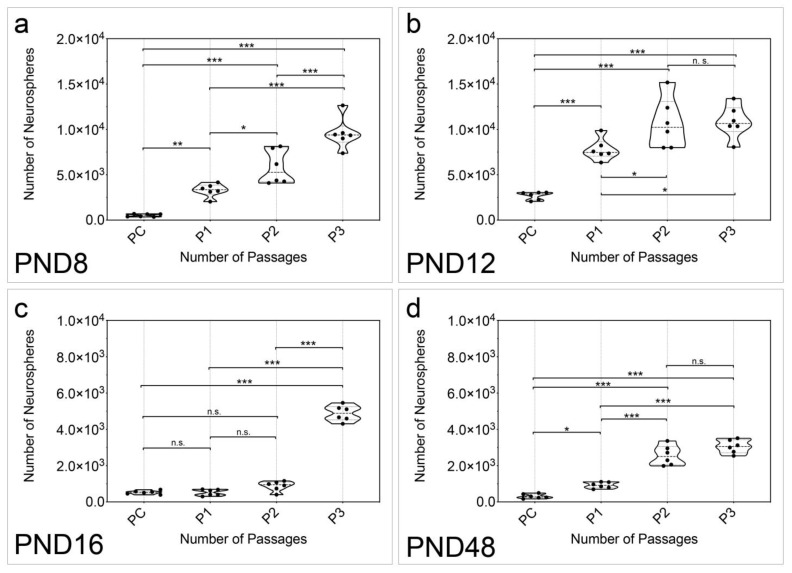
The quantitative examination of the passages showed an increase in the number of neurospheres with increasing passage in all age groups of the MGB. (**a**) The number of neurospheres formed per organ increased significantly between all passages from PC to P3 at PND 8. (**b**) At PND 12, the number of neurospheres per organ increased significantly with increasing passage, except between P2 and P3. (**c**) At PND 16, there was no significant increase between PC, P1, and P2. However, there was a significant increase in neurospheres between these passages and P3. (**d**) At PND 48, the number of neurospheres per organ increased significantly with increasing passage, except between P2 and P3. The violin plots show the median with the upper and lower quartiles; each dot represents a cell culture, n = 6; asterisks indicate the level of significance: n.s. = not significant, * *p* < 0.05, ** *p* < 0.005, *** *p* < 0.001.

**Figure 3 ijms-25-02623-f003:**
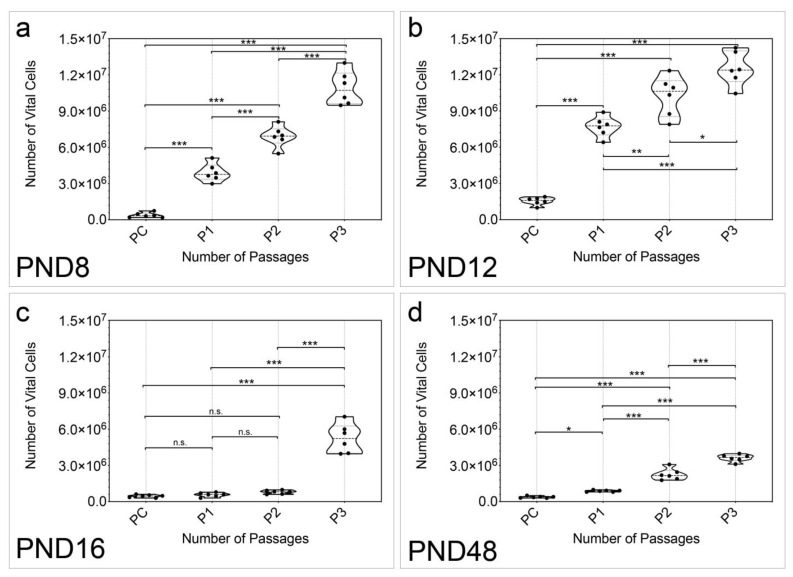
The number of vital cells within the neurospheres increased between PC and P3 at all age groups of the MGB. (**a**,**b**) The number of vital cells within the neurospheres formed increased significantly between all passages from PC to P3 at PND 8 and PND 12. (**c**) At PND 16, there was no significant increase in the number of vital cells between PC, P1, and P2. However, there was a significant increase in between these passages and P3. (**d**) At PND 48, the number of vital cells within neurospheres increased significantly with every increasing passage. The violin plots show the median with the upper and lower quartiles; each dot represents a cell culture, n = 6; asterisks indicate the level of significance: n.s. = not significant, * *p* < 0.05, ** *p* < 0.005, *** *p* < 0.001.

**Figure 4 ijms-25-02623-f004:**
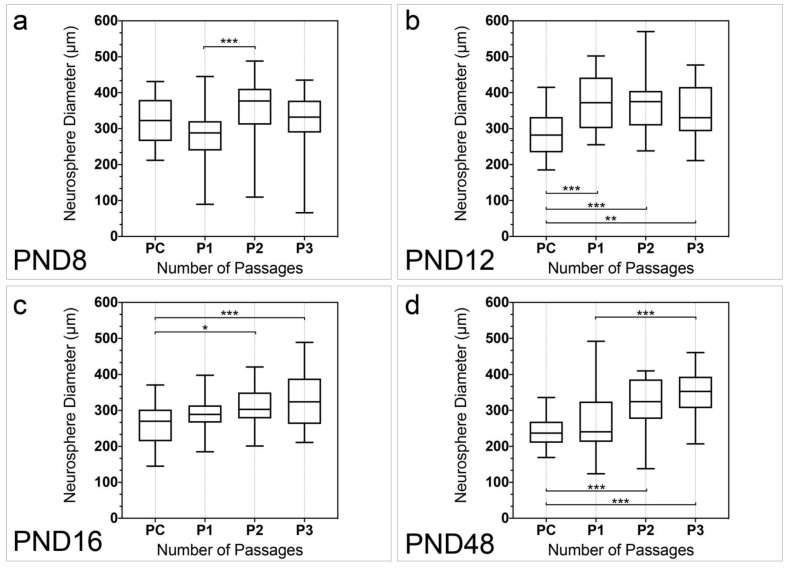
Quantitative analysis of the neurosphere diameter with increasing number of passages from PND 8 to PND 48. (**a**) The diameter of the neurospheres increased significantly between P1 and P2 at PND 8. (**b**) At PND 12, there was a significant increase in the diameter of neurospheres between PC and every following passage. (**c**) The diameter of neurospheres was significantly higher at P2 and P3 in comparison to PC at PND 16. (**d**) At PND 48, the diameter of neurospheres increased from PC to P2 and P3 as well as from P1 to P3. The boxplots show the median with the upper and lower quartiles, and whiskers mark the upper and lower maximum values; per passage: n = 30 (5 neurospheres per cell culture, 6 cell cultures); asterisks indicate the level of significance: * *p* < 0.05, ** *p* < 0.005, *** *p* < 0.001.

**Figure 5 ijms-25-02623-f005:**
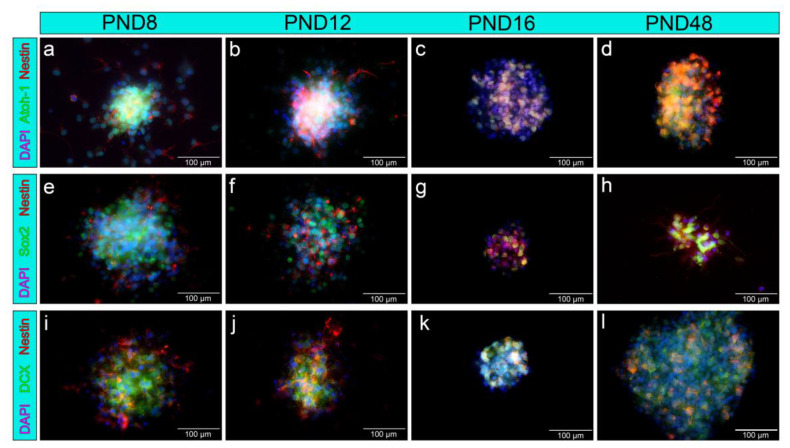
Neurospheres from cells of the rat-MGB-expressed NSC markers up to the adult stage. (**a**–**l**) Neural progenitor cells, emigrating from the neurospheres, stained positively for nestin (red); cell nuclei were stained blue with DAPI. (**a**–**d**) The transcription factor Atoh-1 (green) was positive in the nucleus of cells inside the neurospheres. (**e**–**h**) The nuclei of cells within the neurospheres showed positive labeling for the transcription factor Sox2 (green). (**i**–**l**) The neural progenitor marker doublecortin (DCX, green) was positive in the cytoplasm of emigrating cells and cells within the neurospheres. The immunocytological images of the single antibodies as well as phase contrast microscopic images of the neurospheres shown in this graph are added in the Appendix A.

**Figure 6 ijms-25-02623-f006:**
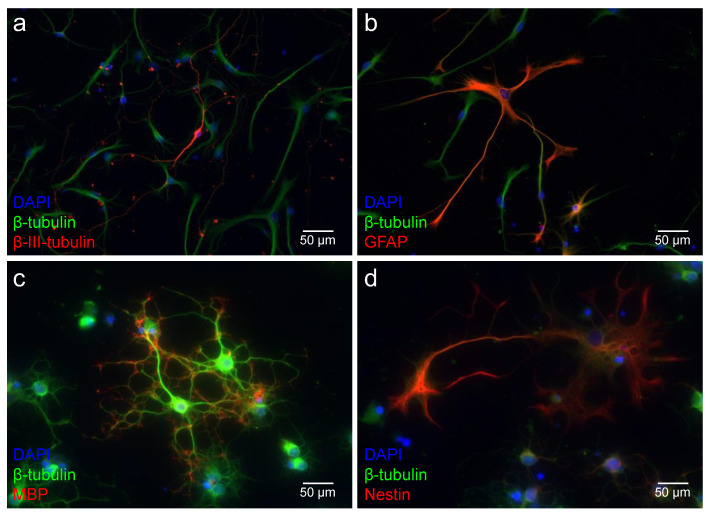
NSCs of the MGB show the capacity to differentiate in all cells of the neuroectodermal lineage up to the adult stage. Single cells of the age groups PND 16 and PND 48 are shown as representative of all age groups. (**a**–**d**) ß-tubulin (green) stained the cytoskeleton of the viable cells, and their nuclei were stained blue with DAPI. (**a**) The proneuronal marker ß-III-tubulin (red) labels neuron-typical cells with their characteristic morphology (PND 16). (**b**) A certain proportion of the differentiated cells displayed positive staining for the astrocytic marker glial fibrillary acidic protein (GFAP, PND 16, red). (**c**) The myelin basic protein identified oligodendrocytes (MBP, PND 48, red). Their peripheral myelinization processes are labeled with MBP. (**d**) The neural progenitor marker nestin (red) marks undifferentiated progenitor cells (PND 48).

**Figure 7 ijms-25-02623-f007:**
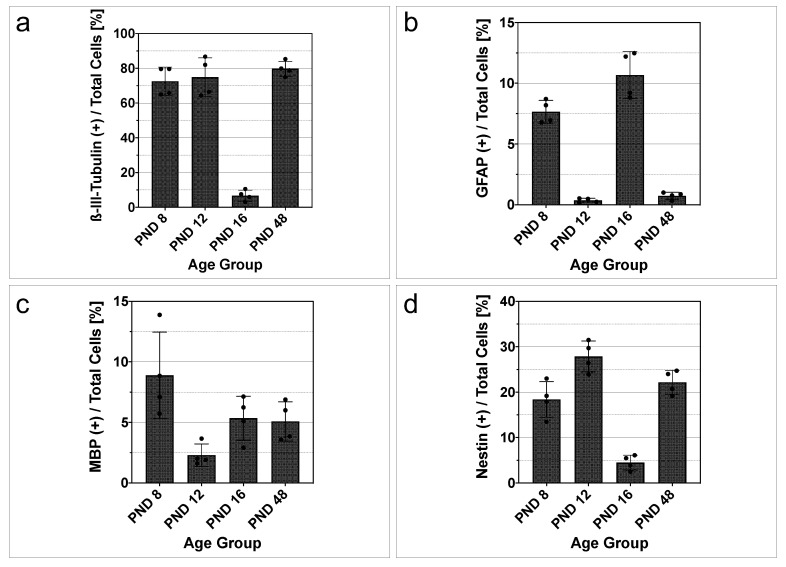
Single cells from rat MGB neurospheres show an age-dependent differentiation capacity. (**a**–**d**) Quantitative investigation of differentiation marker-positive cells in relation to total cells (ß-tubulin) from PND 8 to PND 48 after six days on glass coverslips and differentiation medium. (**a**) At PND 8, PND 12, and PND 48, more than 70% of the cells are positive for ß-III-tubulin, indicating the cells’ proneural fate. At PND 16, the relative expression of ß-III-tubulin is reduced. (**b**) The astrocytic marker glial fibrillary acidic protein (GFAP) shows an increase in relative expression at PND 8 and PND 16. At PND 12 and PND 48, the percentage of GFAP in the total number of cells decreases. (**c**) After a decrease at PND 12, a stable expression of myelin basic protein (MBP) of approximately 5% was detected at PND 16 and PND 48. (**d**) The progenitor marker nestin shows an increase at PND 12 and a decrease at PND 16. At PND 8 and PND 48, the relative expression of nestin was approximately 20%. The bar charts show the mean and standard error of the mean (SEM); each dot represents a cell culture, n = 4.

**Figure 8 ijms-25-02623-f008:**
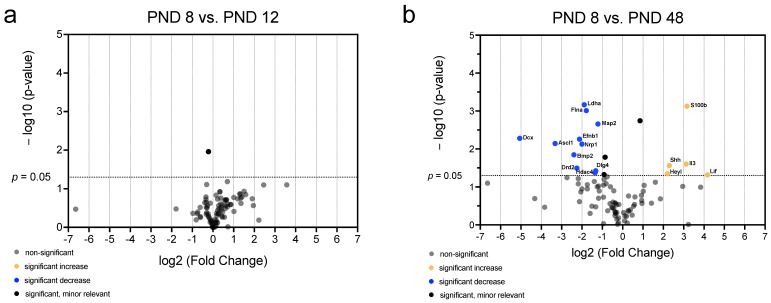
Significant and relevant differences in mRNA abundance of neurogenic factors of the MGB emphasize changes over time up to the adult stage. (**a**) The volcano plot shows no relevant differences in mRNA abundance of neurogenic factors between PND 8 and PND 12. (**b**) The volcano plot highlights significant and relevant differences in mRNA abundance of neurogenic factors between PND 8 and PND 48. The *y*-axis displays the negative logarithm with base 10 of the *p*-value, and the *x*-axis represents the logarithm with base 2 of the fold change. The *p*-value is indicated with the dashed line at the corresponding position. Points below this dashed line have *p* > 0.05. The legend explains the color coding of the individual dots, each representing a gene.

**Figure 9 ijms-25-02623-f009:**
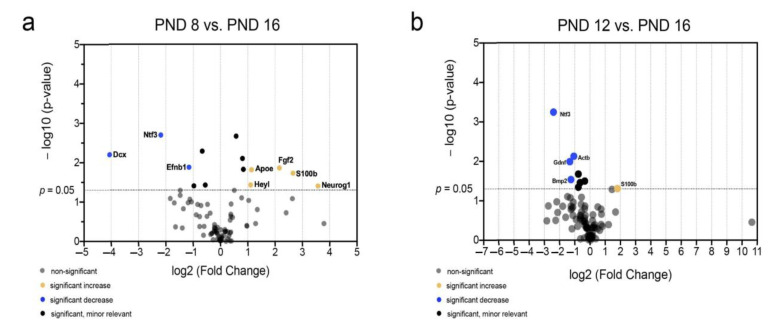
Significant and relevant differences in mRNA abundance of neurogenic factors of the MGB highlight changes around the hearing onset. (**a**) The volcano plot highlights significant and relevant differences in mRNA abundance of neurogenic factors between PND 8 and PND 16. (**b**) The volcano plot highlights significant and relevant differences in mRNA abundance of neurogenic factors between PND 12 and PND 16. The *y*-axis displays the negative logarithm with base 10 of the *p*-value, and the *x*-axis represents the logarithm with base 2 of the fold change. The *p*-value is indicated with the dashed line at the corresponding position. Points below this dashed line have *p* > 0.05. The legend explains the color coding of the individual dots, each representing a gene.

**Figure 10 ijms-25-02623-f010:**
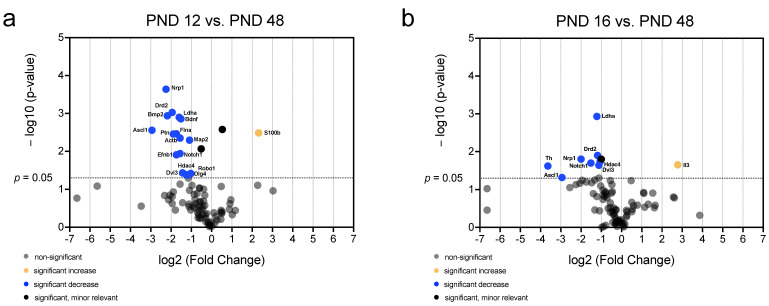
Significant and relevant differences in mRNA abundance of neurogenic factors of the MGB emphasize changes compared to the adult stage. (**a**) The volcano plot highlights significant and relevant differences in mRNA abundance of neurogenic factors between PND 12 and PND 48. (**b**) The volcano plot highlights significant and relevant differences in mRNA abundance of neurogenic factors between PND 16 and PND 48. The *y*-axis displays the negative logarithm with base 10 of the *p*-value and the *x*-axis represents the logarithm with base 2 of the fold change. The *p*-value is indicated with the dashed line at the corresponding position. Points below this dashed line have *p* > 0.05. The legend explains the color coding of the individual dots, each representing a gene.

## Data Availability

The data used to support the findings of this study are included within the article.

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
