# Peer review of "Adult Neurogenesis of the Medial Geniculate Body: In Vitro and Molecular Genetic Analyses Reflect the Neural Stem Cell Capacity of the Rat Auditory Thalamus over Time"

_ijms, 2024, doi:10.3390/ijms25052623_

Round 1

Reviewer 1 Report

Comments and Suggestions for Authors

Author Response

Review #1:

In the manuscript, titled “New Neurons in the Adult Auditory Thalamus: mRNA and Protein Levels Reflect the Neural Stem Cell Capacity of the Rat Medial Geniculate Body over time,”explores the characterization of NSCs in the rat MGB at various ages. The authors found NSCs capable of differentiating into neurons and glial cells across the examined age groups.The authors recently reported the presence of NSCs in the rat MGB at PND8 (Life 2023, 13(5), 1188; https://doi.org/10.3390/life13051188). Consequently, the primary focus of the submittedpaper is on the discovery of NSCs in the adult rat MGB and comparison of properties of NSCs at various ages. Reviewer agrees that if NSCs are found in adult MGBs as well as in auditory cortex and inferior colliculus, it may be novel therapeutic targets of auditory disorders.

Although the manuscript was well written and data are interesting, reviewer asks to authors to solve some problems to ensure the reliability of the paper.

Answer:

The authors would like to thank you very much for the thoughtful review and the recommendations for improvement.

All comments and suggestions for improvement have been carefully reviewed and the authors hope that the ambiguities have been removed and the quality of the work has been improved.

Comment 1

NSCs have different proliferating rate (Figure 2) and differentiating capacity (Figure 5) depending on the ages. Results of immunostaining in Figure 3 show the heterogeneity of NSCs among groups of age, e.g., cells with Atoh-1-immunosignals but not with Nestin-immunosignals are found in spheres at PND8, but hardly found at PND12. Since characterization of NSCs are one of main topic in this paper, reviewer suggests authors to add photos of immunostaining of each antibody with merged photos to provide readers more detailed information about expression of marker proteins. In addition, although RNA profiling does not show any significant differences in Figure 6a and authors suggest close correlations between mRNA and protein levels in Line 403-404,expression level of markers of NSCs seems to be varied between PND8 and PND12 in Figure3as mentioned above. After all, are NSCs at PND8 and PND12 similar or not? Please re-consider the descriptions in Line 392- 395 “This indicates that the mRNA abundance of neurogenic factors of the MGB before (PND8) and around the hearing onset (PND12) is stable and without significant changes at the mRNA level.” and Line 403-404 “These results indicate a closecorrelation between mRNA and protein levels in the proliferative phase before (PND8) and around the hearing onset (PND12)” and revise appropriately.

Discussion about the relations of heterogeneity of expression of markers in NSCs and cell capacity of proliferation and differentiation will help readers to understand the characters of NSCs

in rat MGB.

Answer Comment 1:

Thank you for your review. The improvements will contribute significantly to the quality of the work.

The immunocytological images of the neurospheres in Fig. 3 (Fig. 5 in the revised manuscript) have been completely revised to provide better quality immunosignals. Likewise, the immunosignals of all individual antibodies, the merged images and phase-contrast images in the supplementary material have been created and added as an overview in order to provide the reader with as detailed a presentation as possible.

Regarding the correlation between the mRNA and protein levels between PND 8 and PND 12, the comments you noted (lines 392-395 and 403-405 in the first version of the manuscript) are misleading. Therefore, these have been revised and the discussion has been expanded to include the relationship of heterogeneity of NSC markers.

Comment 2

In Figure 2, authors suggest cells are proliferating because the number of neurospheres is increased after passages. However, authors do not mention anything about the size of spheres. If small spheres, which may be remnant caused by insufficient dispersion of spheres at passages, are generated, increased number of spheres does not necessarily means increased number of cells.

Please add the data of size of spheres in Figure 2.

Answer Comment 2:

Thank you for your review. The improvements contribute significantly to the quality of the work.

If only the number of neurospheres is shown, no conclusions can be drawn about the number of neurosphere diameters and the number of living cells. Therefore, these analyses (Fig. 3 and Fig. 4 in the revised manuscript) were supplemented and included in the results/discussion.

Comment 3

Considering whether this study contribute to the therapies of human disease or not is important points that many readers would like to know. Authors mention that MGB is related to the onset of tinnitus and NSCs in MGB may possibly to contribute to therapy. To consider the application of NSCs in MGB for therapy of tinnitus, the data of aged mice are required because tinnitus associated with presbycusis is unavoidable topics. The latest stages examined in this study is P40,

which is too young even if the auditory system has matured to discuss about tinnitus associated with presbycusis. The reviewer understands the authors used mice with limited age due to the ethical needs to reduce the number of animals used in experiments, however, it is still required to use older mice to discuss about clinical application. Or, is there any other examples of clinical application of your results? Discussions on the potential clinical relevance of the findings in this study is required.

In addition, number of spheres and sphere size at PND40 is much smaller than other spheres as shown Figure 1 and 3. These results make a reviewer worry that NSCs are slowly dying and hardly found in later ages than PND40 in vivo. Do authors have any comments on this issue?

Answer Comment 3:

The authors agree that an association to tinnitus in the context of presbyacusis is limited with the studied age groups of the rat. Therefore, the introduction and parts of the discussion have been revised to emphasise regenerative-translational therapeutic approaches in neurootology, for which the results of this study may provide an important basis. This emphasises the clinical relevance of the findings.

The authors understand very well the reviewer's concern that the impression may arise that NSCs slowly deteriorate with increasing age/passage. Therefore, the immunohistological images were completely revised to present the highest quality images of the immunosignals (including single channel and phase contrast images in the supplementary material). The immunohistological images harbour the risk of drawing conclusions about the size of the neurospheres. In order to reduce this risk and present the results more precisely, the diameters of the neurospheres and the number of individual cells were added to the graphs (Fig. 3 and Fig. 4 in the revised manuscript). There is an increase wit increasing passage in the number of vital cells, very similar to the number of neurospheres, and a less pronounced increase in neurosphere diameter at most ages. This suggests that the NSCs do not die slowly, but have a lower potential with increasing age.

minor points

[1] Removal of horizontal scale lines from the graph in Figure 1 will make it easier to see.

Fig. 1 has been revised to provide a better overview.

[2] Line 91: Since there is no statistically differences between PND12 and PND8, “The highest number of neurospheres was detected at PND12.” is not appropriate description

The description is misleading and has therefore been removed.

[3] Line 138: (m-p) should be replaced by (i-l).

A revision has been made.

[4] Line 270-272: Authors refer the paper published by Shetty et al., (ref no.35) to discuss that very few changes at the adult stage between 2-month-old and 24-month-old adult rats at the mRNA level of neurogenic factors. However, Shetty et al. mainly investigated the dentate gyrus but not NGB or whole brain. Please replace with more appropriate references or remove the

description.

In this context, the reference is not very meaningful and potentially misleading and has therefore been removed.

[5] Line 284: FGFb should be replaced with bFGF of FGF2.

A revision has been made.

[6] Line 334: Reviewer does not understand well the meanings of “strains the MGB NSC niche.”Do authors mean auditory signals stimulate the NSCs to differentiate into neurons and/or glial cell after hearing onset? Is there any previous study which showed that the basic sensory stimulation decreases the number of NSCs in vivo rather than injuries or diseases (ref. no 46 and 47)?

Line 720: “suggesting a high stimulation of the NSC niche prior” is hard to understand. Reviewer understand that stimulation of NSC niche by sound is one of the possibilities. Still, there is another possibility, e. g., NSCs are simply dying because of a certain changes in circumstances?

The wording is ambiguous. It has therefore been revised and references relating to pathological influences have been replaced by references relating to the influence of sensory stimulation.

Furthermore, the discussion was revised and a paragraph on dynamics of the NSC/cell culture model was added, which takes into account certain changes under the influence of the NSC medium.

[7] Line 530: PND 40+ means what? Did authors use the mixture of rat at different ages around PND40? Please specify the age of rats used in this study.

As explained in the manuscript, the authors had based their choice of age groups on the maturation of the auditory pathway. Therefore, too little precision was used in the specification of the "PND 40+/adult" animals. The animals are PND 48 in the oldest age group. The manuscript has been completely revised accordingly (text and graphics).

[8] Line 591-592: Authors mentioned that plates were coated with poly-D-lysine and laminin. Were plates coated with mixture of poly-D-lysine and laminin, or in sequential manner? Information of period and temperature for coating is also required

The relevant paragraph has been supplemented with the necessary information and revised.

Reviewer 2 Report

Comments and Suggestions for Authors

The authors conducted an investigation into the presence of neural stem cells in the rat medial geniculate body (MGB). Cells from the MGB region of rats at various ages were isolated, and their neural stem cell and differentiation capacities were analyzed through the examination of neuron markers at both protein and mRNA levels. However, the novelty of the findings is unclear, as the existence of adult stem cells in organs is already established, with their numbers known to decrease with aging. The authors need to provide a more compelling rationale for the uniqueness and significance of their study.

Furthermore, the attempt to associate neural stem cells with the auditory system is questionable, given that a 7-day differentiation protocol was employed to generate general neurons rather than specific neurons. Additional comments include:

1)      Inclusion of phase-contrast images of neurospheres in the results would enhance clarity.

2)      The number of starting cells for passaging analysis should be added.

3)      Phase contrast and non-merged images for all figures should be included in the supplementary file.

4)      Clarification is needed on the use of "cell line" in line 143. Did the authors aim to derive cell lines or obtain differentiated neuron cells?

5)      In Figure 5, the authors should explain how different neurons were obtained using the same differentiation protocol, especially for cell lineages like oligodendrocytes that typically require a longer in vitro differentiation time.

6)      Line 162 is ambiguous and requires rewriting for clarity.

7)      Line 163 should provide detailed information on how the positive percentage cells were determined, including the number of replicates and fields.

8)      The decrease in the ß-III-tubulin marker after differentiation needs further explanation, as it is unexpected for this marker to disappear after in vitro differentiation. While discussed in the paper, more clarity is required.

9)      Results for PND16 cells after differentiation are unusual, and the authors should investigate further to determine the fate of these cells, considering various neuron markers expressed in all neurons.

10)   Line 194 mentions "< 0.05"; clarification is needed if "< 0.5" was intended.

11)   Line 460 should specify whether "neuritogenesis" or "neurogenesis" is meant.

12)   The discussion is lengthy and challenging to follow. To improve readability, consider adding conclusions based on expression data at the end of each results section. Reserve the discussion for other aspects of the data. 

Comments on the Quality of English Language

The English needs minor revisions

Author Response

Review #2:

  1. The authors conducted an investigation into the presence of neural stem cells in the rat medial geniculate body (MGB). Cells from the MGB region of rats at various ages were isolated, and their neural stem cell and differentiation capacities were analyzed through the examination of neuron markers at both protein and mRNA levels. However, the novelty of the findings is unclear, as the existence of adult stem cells in organs is already established, with their numbers known to decrease with aging. The authors need to provide a more compelling rationale for the uniqueness and significance of their study.

Answer:

Thank you for your thorough review. The authors agree that the existence of adult neural stem cells (NSCs) is known and that a decreasing capacity with age is already recognised in different brain regions. However, it is unknown whether the medial geniculate body (MGB) possesses adult NSC potential. There parts of the auditory pathway that only have neonatal and not adult NSC potential. Explicitly for the auditory pathway, the question arises as to how pronounced the NSC potential of individual nuclei is over time, since, for example, the ability to form spheres is lost in the cochlea after the third postnatal week (Oshima et al. 2007). The NSC potential in the adult cochlear nucleus is also very low (Rak et al. 2013). Thus, a persistent, though decreasing, adult NSC potential in the auditory pathway is of significance. It is of great importance to know up to what age and how pronounced the NSC potential is in order to assess a basis for a regenerative approach. Investigations of how NSC niches age are of crucial importance for the treatment of neurodegenerative diseases. Clinically, this is increasingly important and in otology there are only rehabilitative, but no regenerative options against sensorineural hearing loss (Waqas et al. 2020). NSCs therefore represent an attractive basis for regenerative medicine.

The translational character and the significance of this study were not sufficiently described in the initial version and are now emphasised in detail in the revised version. The revisions in response to your incentive represent a significant improvement in the quality of the work.

  1. Furthermore, the attempt to associate neural stem cells with the auditory system is questionable, given that a 7-day differentiation protocol was employed to generate general neurons rather than specific neurons.

Answer:

Thank you for your thorough review. The authors agree that no specific neurons were detected in the studies. The aim of the study was rather to demonstrate the characteristics of NSC in in-vitro experiments at different ages of the MGB, as the existence of NSC in the adult MGB is not known.  Furthermore, it should be analysed how the mRNA levels of factors with high relevance for NSCs and neurogenesis behave over time. Thus, the basis for further analyses up to the adult stage should be established and it should be determined whether this potential can also be represented at the mRNA level. Of course, it is of great interest whether the cells generated from NSC of the MGB are able to generate specific neurons. In addition, functional investigations such as the patch-clamp technique would be necessary. These investigations, which build on this project, are of interest for the further analysis of regenerative capacity, but were not part of this project, as the basic aim was to find out whether the MGB is home to adult neurogenesis/adult NSCs at all and how the mRNA levels of neurogenetic factors develops. In order to take these important points into account, this was included in the limitations of the study and serves as an outlook for the subsequent projects. This note increases the quality of the work.

In order to avoid confusion for the reader, the title of the manuscript has been adapted accordingly.

Additional comments include:

1)      Inclusion of phase-contrast images of neurospheres in the results would enhance clarity.

Thank you for your thorough review. The quality of the work has been significantly improved. All immunocytological images of the neurospheres in Fig. 3 (Fig. 5 in the revised manuscript) have been revised and phase-contrast images have been prepared and are published in the supplementary materials.

.2)      The number of starting cells for passaging analysis should be added.

Thank you for your thorough review. The quality of the work has been significantly improved. The number of vital cells at all ages and passages was added and shown in Fig. 3 (in the revised manuscript).

3)      Phase contrast and non-merged images for all figures should be included in the supplementary file.

Thank you for your thorough review. The quality of the work has been significantly improved. All immunocytological images (single and merged) of the neurospheres in Fig. 3 (Fig. 5 in the revised manuscript) have been revised and phase-contrast images have been prepared and are published in the supplementary materials.

4)      Clarification is needed on the use of "cell line" in line 143. Did the authors aim to derive cell lines or obtain differentiated neuron cells?

Thank you for your thorough review. The wording is misleading and has therefore been revised.

5)      In Figure 5, the authors should explain how different neurons were obtained using the same differentiation protocol, especially for cell lineages like oligodendrocytes that typically require a longer in vitro differentiation time.

Thank you for your thorough review. In this study, single cells on glass covers were first kept in NSC medium for 3 days and then in differentiation medium for 7 days. There is an inaccuracy in the description of the methods. A corresponding revision was made in 4.2. Progenitor (Nestin), glial cells (astrocytes/GFAP, oligodendrocytes/MBP) and neurons were cultured. The time span corresponds to empirical values that have developed from the work of our research group with NSCs of the auditory pathway. In our experience, oligodendrocytes need longer to differentiate than neurons and astrocytes. However, some research groups were able to generate MBP-positive oligodendrocytes in culture out of Oligodendrocyte Progenitor Cells (OPC) after just 5 days (Janowska, Ziemka-Nalecz, and Sypecka 2018). One reason for the rapid differentiation of oligodendrocytes in this study could also be that OPCs already form in the neurospheres of the NSC medium. The differentiation of oligodendrocytes from neurosphere cells enables rapid generation. OPCs were generated from rodent neurospheres within 4-6 days and oligodendrocytes from these in a further 3 days (Li et al. 2019). The density at which the cells are plated also plays a decisive role in the differentiation of oligodendrocytes. In our experiments, 100/mm2 (10,000/cm2) cells were plated. It was shown that significantly more differentiation takes place at low cell numbers (15,000/cm2) compared to high cell numbers (50,000/cm2) (Janowska, Ziemka-Nalecz, and Sypecka 2018). The authors consider these factors to be decisive for the success of cultivating oligodendrocytes.

6)      Line 162 is ambiguous and requires rewriting for clarity.

Thank you for your review. The relevant passage was ambiguous and has been revised.

7)      Line 163 should provide detailed information on how the positive percentage cells were determined, including the number of replicates and fields.

Thank you for your thorough review. The relevant additions have been added. 4 glass cover slips from 4 different cultures were analysed completely (entire glass cover slip, divided into 36 identical fields) by 3 experienced investigators each.

8)      The decrease in the ß-III-tubulin marker after differentiation needs further explanation, as it is unexpected for this marker to disappear after in vitro differentiation. While discussed in the paper, more clarity is required.

Thank you for your thorough review. The improvements contribute significantly to the quality of the work. The discussion has been revised and the results of ß-III-tubulin have been discussed in more detail.

9)      Results for PND16 cells after differentiation are unusual, and the authors should investigate further to determine the fate of these cells, considering various neuron markers expressed in all neurons.

Thank you for your review. The authors agree that the results of the differentiation tests in PND 16 particularly stand out. Therefore, the analyses in PND 16 have already been performed in duplicate. Additionally, it was therefore discussed again under 3.2. that different factors must be taken into account when evaluating the ß-III tubulin analyses in different age groups and in in vitro tests. It is possible that a specialised, non-ß-III-tubulin positive neuronal lineage is increasingly expressed in PND 16. However, since the aim of this study was not to differentiate and generate specific neurones, astrocytes or oligodendrocytes of the MGB, but to determine whether a basis for regenerative therapy approaches such as NSC modification is even present in the MGB, a proneural marker was selected rather than several proneural markers.

The function, stimulation and significance of NSC niches differ from the local conditions (Ruddy and Morshead 2018). Therefore, a profound and precise understanding of the NSC niche, which includes the molecular genetic level, is of great importance. NSC niches are very heterogeneous and the formation and differentiation to new neurons is only one possible function of the NSC niche (Andreotti et al. 2019). The communication with its microenvironment and the secretion of stimulating, proliferating and protective factors is another potential function of an NSC niche (Fuentealba, Obernier, and Alvarez-Buylla 2012). Differentiation into neurons is one of several potentially regenerative functions of the NSC niche.

Of course, it is of great interest whether the neurons, and even the glial cells, differentiated from NSC of the MGB are able to become different and specific cells and which cell lineage they develop. In addition, functional investigations such as the patch clamp technique would be necessary. These investigations, which build on this project, are of interest for the further analysis of regenerative capacity, but were not part of this project, as the basic aim was to find out whether the MGB harbours adult neurogenesis/adult NSCs, how the mRNA levels of neurogenetic factors develop in the ageing process and which neurogenic factors play an essential role in MGB Neurogenesis over time. In order to take these important points into account, this was included in the limitations of the study and serves as an outlook for the subsequent projects.

10)   Line 194 mentions "< 0.05"; clarification is needed if "< 0.5" was intended.

Thank you for your thorough review. A revision has been made (0.5 was intended).

11)   Line 460 should specify whether "neuritogenesis" or "neurogenesis" is meant.

Thank you for your thorough review. The relevant passage has been checked. Il3 plays an important role in neurite outgrowth. The relevant literature (Kamegai et al., 1990) is attached in the manuscript.

12)   The discussion is lengthy and challenging to follow. To improve readability, consider adding conclusions based on expression data at the end of each results section. Reserve the discussion for other aspects of the data.

Thank you very much for your careful review. The revisions clearly contribute to improving the quality of the paper. The authors understand the reviewer's concern that the discussion is too long. Short summaries have been placed at the end of each results section of the expression studies and the discussion has been shortened and revised.

Literatur

Andreotti, J. P., W. N. Silva, A. C. Costa, C. C. Picoli, F. C. O. Bitencourt, L. M. C. Coimbra-Campos, R. R. Resende, L. A. V. Magno, M. A. Romano-Silva, A. Mintz, and A. Birbrair. 2019. 'Neural stem cell niche heterogeneity', Semin Cell Dev Biol, 95: 42-53.

Fuentealba, L. C., K. Obernier, and A. Alvarez-Buylla. 2012. 'Adult neural stem cells bridge their niche', Cell Stem Cell, 10: 698-708.

Janowska, J., M. Ziemka-Nalecz, and J. Sypecka. 2018. 'The Differentiation of Rat Oligodendroglial Cells Is Highly Influenced by the Oxygen Tension: In Vitro Model Mimicking Physiologically Normoxic Conditions', Int J Mol Sci, 19.

Li, S., J. Zheng, L. Chai, M. Lin, R. Zeng, J. Lu, and J. Bian. 2019. 'Rapid and Efficient Differentiation of Rodent Neural Stem Cells into Oligodendrocyte Progenitor Cells', Dev Neurosci, 41: 79-93.

Oshima, K., C. M. Grimm, C. E. Corrales, P. Senn, R. Martinez Monedero, G. S. Geleoc, A. Edge, J. R. Holt, and S. Heller. 2007. 'Differential distribution of stem cells in the auditory and vestibular organs of the inner ear', J Assoc Res Otolaryngol, 8: 18-31.

Rak, K., J. Volker, S. Frenz, A. Scherzed, A. Radeloff, R. Hagen, and R. Mlynski. 2013. 'Dynamic changes of the neurogenic potential in the rat cochlear nucleus during post-natal development', Exp Brain Res, 226: 393-406.

Ruddy, R. M., and C. M. Morshead. 2018. 'Home sweet home: the neural stem cell niche throughout development and after injury', Cell Tissue Res, 371: 125-41.

Waqas, M., I. Us-Salam, Z. Bibi, Y. Wang, H. Li, Z. Zhu, and S. He. 2020. 'Stem Cell-Based Therapeutic Approaches to Restore Sensorineural Hearing Loss in Mammals', Neural Plast, 2020: 8829660.

Round 2

Reviewer 1 Report

Comments and Suggestions for Authors

The authors have satisfactorily addressed my concerns, and the manuscript has seen significant improvement. 

Allow me to highlight two minor points that could further enhance the quality of the manuscript:

[1] In my previous comments, I noted that removing some horizontal scale lines from the graph in Figure 1 would enhance visibility. Could the authors consider reducing the number of horizontal scale lines in Figures 1-4 to emphasize the actual values?

[2] The reviewer has pointed out that the word “auf” in line 553 may not be an English word. Do the authors intend to use the word “of” instead?"

Author Response

The authors have satisfactorily addressed my concerns, and the manuscript has seen significant improvement.

Answer:

Thank you again for your accurate review. The improvements have contributed significantly to the quality of the manuscript.

Allow me to highlight two minor points that could further enhance the quality of the manuscript:

[1] In my previous comments, I noted that removing some horizontal scale lines from the graph in Figure 1 would enhance visibility. Could the authors consider reducing the number of horizontal scale lines in Figures 1-4 to emphasize the actual values?

Answer:

Thank you for your suggestion for improvement. The horizontal scaling lines in Fig. 1- Fig. 4 have been reduced to emphasise the values.

[2] The reviewer has pointed out that the word “auf” in line 553 may not be an English word. Do the authors intend to use the word “of” instead?"

Answer:

Thank you for your careful review. The wording has been corrected.

Reviewer 2 Report

Comments and Suggestions for Authors

The authors have addressed all necessary revisions. I have no further comments.

Author Response

Thank you again for your accurate review. The improvements have contributed significantly to the quality of the manuscript.